# Subsurface microbial community structure shifts along the geological features of the Central American Volcanic Arc

**Marco Basili**[1,2,3], **Timothy J. Rogers**[1], **Mayuko Nakagawa**[4], **Mustafa Yücel**[5], **J. Maarten de Moor**[6,7], **Peter H. Barry**[8], **Matthew O. Schrenk**[9], **Gerhard L. Jessen**[10,11], **Ricardo Sánchez-Murillo**[12], **Sabin Zahirovic**[13], **David V. Bekaert**[8,14], **Carlos J. Ramirez**[15], **Deborah Bastoni**[2], **Angelina Cordone**[2], **Karen G. Lloyd**[1,16]*, **Donato Giovannelli**[2,3,4,8,17]*

1 Department of Microbiology, University of Tennessee, Knoxville, TN, United States of America, 2 Department of Biology, University of Naples "Federico II", Naples, Italy, 3 Institute for Marine Biological Resources and Biotechnologies, National Research Council (CNR-IRBIM), Ancona, Italy, 4 Earth-Life Science Institute, Tokyo Institute of Technology, Tokyo, Japan, 5 Institute of Marine Sciences, Middle East Technical University, Mersin, Turkey, 6 OVSICORI, Universidad Nacional, Heredia, Costa Rica, 7 Department of Earth and Planetary Sciences, University of New Mexico, Albuquerque, New Mexico, United States of America, 8 Marine Chemistry & Geochemistry Department, Woods Hole Oceanographic Institution, Woods Hole, MA, United States of America, 9 Department of Earth and Environmental Sciences, Department of Microbiology and Molecular Genetics, Michigan State University, Lansing, MI, United States of America, 10 Instituto de Ciencias Marinas y Limnológicas, Universidad Austral de Chile, Valdivia, Chile, 11 Center for Oceanographic Research COPAS COASTAL, Universidad de Concepción, Concepción, Chile, 12 Department of Earth and Environmental Sciences, Tracer Hydrology Group, University of Texas, Arlington, TX, United States of America, 13 School of Geosciences, The University of Sydney, Darlington, Australia, 14 CRPG, Vandœuvre-lès-Nancy, France, 15 Servicio Geológico Ambiental (SeGeoAm), Heredia, Costa Rica, 16 Earth Science Department, University of Southern California, Los Angeles, CA, United States of America, 17 Department of Marine and Coastal Science, Rutgers University, New Brunswick, NJ, United States of America

* donato.giovannelli@unina.it (DG); lloydk@usc.edu (KGL)

**Data Availability Statement:** All the sequences analyzed in this study are available through NCBI under project PRJNA797441 and ENA under

## Abstract

Subduction of the Cocos and Nazca oceanic plates beneath the Caribbean plate drives the upward movement of deep fluids enriched in carbon, nitrogen, sulfur, and iron along the Central American Volcanic Arc (CAVA). These compounds fuel diverse subsurface microbial communities that in turn alter the distribution, redox state, and isotopic composition of these compounds. Microbial community structure and functions vary according to deep fluid delivery across the arc, but less is known about how microbial communities differ along the axis of a convergent margin as geological features (*e.g.*, extent of volcanism and subduction geometry) shift. Here, we investigate changes in bacterial 16S rRNA gene amplicons and geochemical analysis of deeply-sourced seeps along the southern CAVA, where subduction of the Cocos Ridge alters the geological setting. We find shifts in community composition along the convergent margin, with communities in similar geological settings clustering together independently of the proximity of sample sites. Microbial community composition correlates with geological variables such as host rock type, maturity of hydrothermal fluid and slab depth along different segments of the CAVA. This reveals tight coupling between deep Earth processes and subsurface microbial activity, controlling community distribution, structure and composition along a convergent margin.

project accession PRJEB63479. A complete R script containing all the steps to reproduce our analysis is available at https://github.com/giovannellilab/Basili_et_al_Central_America_Convergent_Margin with DOI https://zenodo.org/doi/10.5281/zenodo.10578391 together with all the environmental and geochemical data.

**Funding:** The authors acknowledge the Biology Meets Subduction Project, funded by the Alfred P. Sloan Foundation and the Deep Carbon Observatory (G-2016-7206) to P.H.B, J.M.d.M, D. G., and K.G.L, with DNA sequencing from the Census of Deep Life. Additional support came from NSF FRES (Award# 21211637) to P.H.B., J.M.d.M and K.G.L, NSF Award OCE-2151015 to P.H.B. and K.G.L.. U. S. Department of Energy, Office of Science, Office of Biological and Environmental Research, Genomic Science Program (DE-SC0020369 to K.G.L). FONDECYT Grant 11191138 and COPAS COASTAL ANID FB210021 (ANID Chile) to G.L.J. D.G. was partially supported by funding from the European Research Council (ERC) under the European Union's Horizon 2020 research and innovation program Grant Agreement No. 948972—COEVOLVE—ERC-2020-STG. M.B. was funded by the EU CampusWorld scholarship from UNIVPM to visit the laboratory of K.G.L. in the framework of a research collaboration between K. G.L. and D.G. S.Z. was supported by Australian Research Council grant DE210100084, and Alfred P Sloan grants G-2017-9997 and G-2018-11296. GPlates development is funded by the AuScope National Collaborative Research Infrastructure System (NCRIS) program.

## Introduction

Subduction zones are the primary tectonic settings that transfer volatiles between Earth's surface and subsurface [1,2]. Many of these volatiles, such as inorganic carbon and redox-active elements like carbon, nitrogen, sulfur, and iron compounds, are biologically reactive and can be used for biomass synthesis and energy production. Earth's subsurface harbors a vast microbial community limited by the depth of the 122°C isotherm [3,4]. However, our understanding of how deep life interacts with the heterogeneous distribution of volatiles fluxing through subduction zones is limited. Recently, the subsurface microbial community of a ~400 km subduction segment traversing the Central American Volcanic Arc (CAVA) has been shown to shift composition and metabolic properties in response to across-arc variation in fluxes of slab- and mantle-derived volatiles [5,6]. However, the geological setting can also vary *along* the axis of a convergent margin, because of changes in the nature of the downgoing slab vs. the overriding plate, or other secondary features such as ridges or seamounts. Across-arc refers to moving from the trench, to the outer forearc, to the forearc, to the arc, while along-arc refers to moving into different sections of the convergent margin along its axis (Fig 1 arrows). Currently, it is unknown whether such along-axis geological variation drives changes in the microbial community.

Previous across-arc work shows that subsurface chemosynthesis-based communities vary by fluid sources and upper plate processes (e.g., [6,8]). Similar ecosystems have been observed in underwater mud volcanoes, where the delivery of magmatic gasses supports well-developed ecosystems [9]. At serpentinizing ophiolite systems, the geological setting drives variation in the subsurface microbial community through differential volatile deliveries [10]. The conversion of carbon into biomass by these chemosynthetic communities may even be sufficient to impact the overall carbon budget of the subducting margin [5,8], with implications for our understanding of the global carbon cycle over Earth's history.

Unique subsurface microbial communities have also been identified in the backarc of the Izu-Bonin subduction zone [11] and Taupō Volcanic Zone [12], as well as the forearc of the Mariana convergent margin [13], the Sunda subduction zone [14] and the Peru convergent margin [15]. A large-scale backarc study [12] determined that microbial diversity is primarily influenced by pH at temperatures < 70°C, with the effect of temperature becoming more significant for temperatures > 70°C. This study found an increase in community dissimilarities with distance between sites, with niche selection driving assembly on a local scale. Previous studies found changes in microbial communities across partial along-arc sections convergent margins of Costa Rica and Peru [5,6,15].

Costa Rica and Panama have complex geological settings due to the intersection of several tectonic plates (Fig 1A; Caribbean, Cocos, Nazca, and the Panama microplate) and the Cocos Ridge [7]. This complex tectonic setting creates regional-scale along-arc differences in volatile fluxes. The most prominent tectonic features in the area are: a) the Middle American Trench; b) the Cocos Ridge, a submarine volcanic range formed by the Galapagos hot spot track that is subducted in southern Costa Rica; and c) the Panama Fracture Zone, a transform fault zone that defines the triple junction between the Cocos, Nazca, and Caribbean plates off the coast of southern Costa Rica and northern Panama. Costa Rica is traversed by four mountain ranges, with most volcanoes located in the Guanacaste and Central volcanic ranges [16,17]. Central-Northern Costa Rica is characterized by basaltic-andesitic volcanism with multiple volcanic features such as thermal waters, acidic rivers or streams rich in sulfur, iron, and silicates [16]. In contrast to Costa Rica, Panama is characterized by a major left-lateral transform fault, along which the Nazca Plate is moving eastwards and is subducting beneath Colombia. This results in a slab window, where a break in the subducting slab allows for a greater mantle influence in

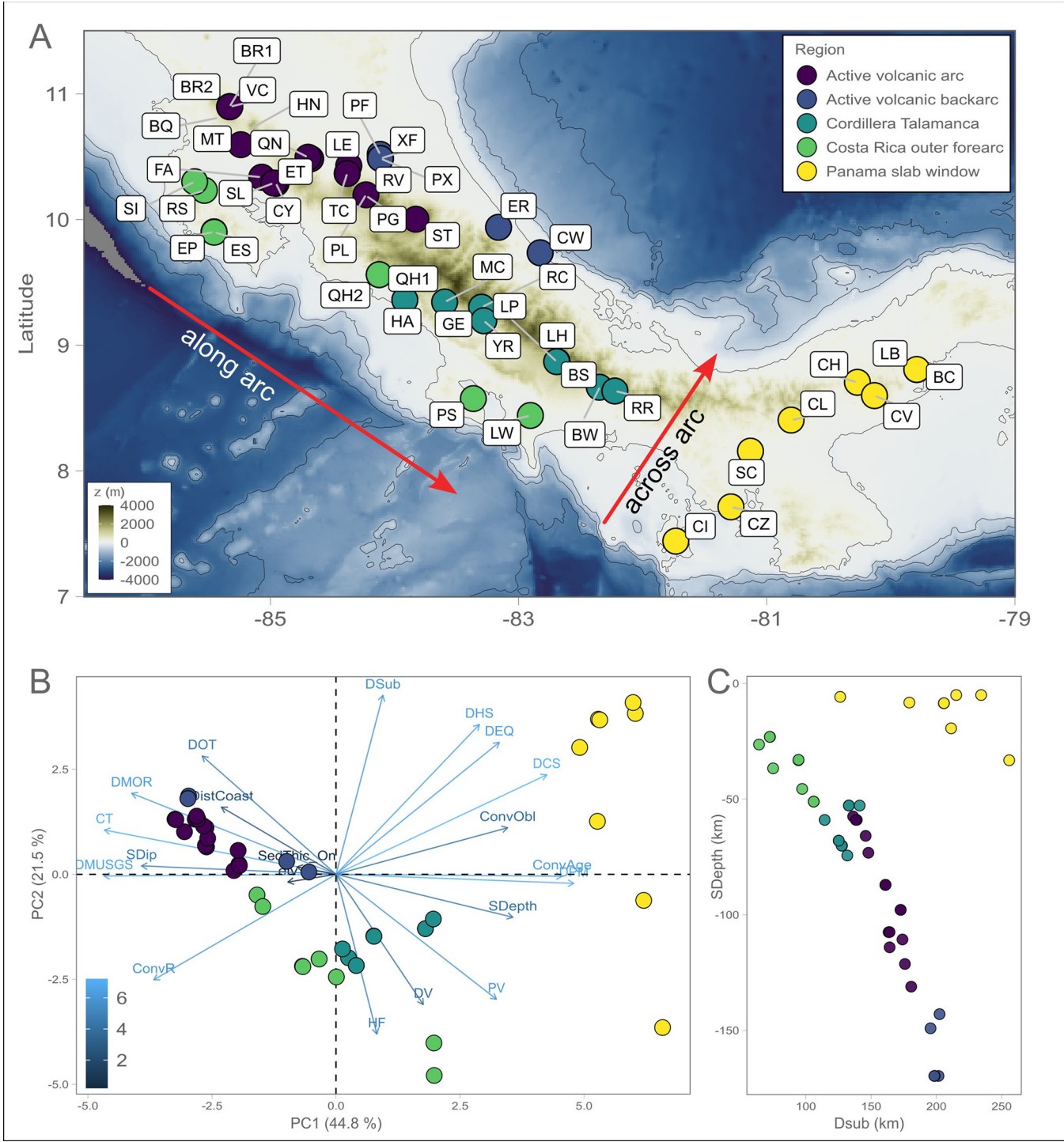

**Fig 1. Map and the sampled sites divided in regions and geophysical parameters defining each region. A** Locations of the sampled seeps along the Central American convergent margin, from Costa Rica and Panama, depicting the bathymetry and the different plates that make up the region. Sites are colored by region. **B** Principal component analysis (PCA) showing the sites' clustering based on their geophysical characteristics. Vectors show the direction of change of the variable in the ordination space and the transparency of the vector is in accordance with the contribution to the corresponding orthogonal PCA axis. The percentage of variance explained is reported for each axis. DOT = Distance from oceanic transform, DistCoast = Distance from coastline, DMOR = Distance from Mid Ocean Ridge, CT = Crustal Thickness, SDip = Slab dip angle, DMUSGS = Distance from major faults, ConvR = Convergence rate, HF = heat flow, DV = Distance from volcanoes, PV = Plate velocities, SDepth = Slab depth, DPM = Distance from passive margin, ConvObl = Convergence obliquity, DCS = Continental shelf, DEQ = Distance from significant earthquake epicenter, DHS = Distance from hot spots, and Dsub = Distance from subduction trench. **C** Relationship between the slab depth (SDepth in km) and the distance from the trench (DSub in km) for each of the sample sites showing the presence

of a slab window under the Panama sites [7]. Arrows show what is meant by "across-arc" and "along-arc". Imagery reproduced from the GEBCO_2022 Grid, GEBCO Compilation Group (2022) GEBCO 2022 Grid (doi:10.5285/e0f0bb80-ab44-2739-e053-6c86abc0289c).

the area [7]. The occurrence of this slab window notably explains the absence of arc volcanism and the relatively low seismic activity in Western Panama [7,18].

Within the CAVA, only a handful of hot springs have been microbiologically characterized [5–7,10–21]. We sampled deeply-sourced fluids from natural springs and wells to investigate whether their microbial compositions differ by regional tectonic province across the CAVA. In oceanic settings, warm fluids emanating from hydrothermal vents can be used as windows to peer into the rocky subseafloor habitat and characterize its microbial community [22]. Similarly, sampling freshly expressed fluids before they pool at the surface in terrestrial hot springs provides access to subsurface microbial communities that are flushed out by the deeply-derived fluids [5,6,15,23,24]. Previous work has shown that common soil microbes and laboratory contaminants comprise only minority populations in these samples and do not correlate with the tracers of deeply-derived geochemical variables [5,6,23]. Furthermore, deep fluid communities, which have been shown to be from deep sources because they have relatively high $^3$He/$^4$He suggesting significant contributions from mantle volatiles and other geochemical indicators of active hydrothermal activity (e.g., hydrothermally derived anions and cations) [8,23–25], do not overlap with the microbial communities in the surrounding soils [24]. Here, we compare changes in subsurface microbial communities along a ~700 km section of the CAVA spanning Costa Rica and Panama. Our results show that microbial community composition shifts significantly according to geological changes along the CAVA, supporting the fundamental role that tectonic processes play in shaping subsurface microbial ecosystems which in turn affect the composition and quantity of volatiles recycled between Earth's interior and its surface.

## Material and methods

### Site description and sampling

During two sampling campaigns in 2017 and 2018, 48 deeply-sourced seeps were sampled across Costa Rica and Panama in the Central American Volcanic Arc (CAVA) (Fig 1A, S1 Table in S1 File) following a large-scale approach [23]. Additional information regarding the geology of the CAVA are provided in the Supplementary Online Materials. From each site, 0.5 to 2 liters of hydrothermal fluids venting from the subsurface and ~ 15 grams of nearby sediments were collected. All samples were natural springs except for the following artificial wells (La Estrella LE, Casa Valmor CV, PraxAir Well PX, Cahuita Well CW, Coiba Island CI, Bajo Mendes Well BW, and Laurel Well LW). Fluids were immediately filtered through Sterivex 0.22 µm filter cartridges (MilliporeSigma) and quick-frozen onsite in liquid nitrogen, along with sediments. Fluid samples were collected for trace metals, major ions, cations and dissolved inorganic carbon (DIC), as described in Fullerton et al [5]. These were combined with samples for gas composition and noble gas analyses as reported previously [5,8]. The authors declare that all biological samples were collected under the authorized 2014 permission given to RS from the Comisión Nacional para la Gestión de la Diversidad de Costa Rica, Ministerio de Ambiente y Energía.

### Aqueous, sediment and rock geochemistry

Detailed methods for geochemical measurements and limits of detection were previously reported [5]. Briefly, concentrations of anions and cations were determined via ion

chromatography (Dionex AS4A-SC column, sodium hydroxide eluent and ASRS-I suppressor for anions and Dionex CS12-SC column, with methane sulfonic acid eluent and CSRS-I suppressor for cations. Two factors were obtained based on the ternary plot classification of the aqueous geochemical composition (cations and anions [26]) of the fluids at each site (Fig 2). The bulk composition of hosting rocks was derived from EarthChem (https://www.earthchem.org/) [27] by querying the database with the coordinate of each site. The obtained whole rock compositions were checked against geological maps of the area to confirm the provenance from the same geologic unit and rock type hosting the samples seeps. Rock geochemistry data were used to investigate the correlation with host rock type.

## Noble gas geochemistry

Noble gas analyses were conducted in various labs, as described in Barry et al [8] and Bekaert et al [7], where the data were originally published.

   **Geophysical data.** We derived the following data from GPlates [29]: crustal thickness (CT), distance from oceanic transformation (DOT), slab dip (S.Dip), distance from earthquakes (DEQ), distance from hotspot (DHS), slab depth (S.Depth), convergence rate (ConvR), Convergence Obliquity (ConvObl), convergence age (ConvAge), distance from mid oceanic ridge (DMOR), distance from nearest volcano (DV), distance from nearest fault identified by the U. S. Geological Survey (DMUSGS), distance from earthquakes (DEQ), and distance from the continental shelf (DCS).

## Community DNA extractions

DNA was extracted from Sterivex® filters and sediments using a modified phenol-chloroform extraction optimized for low biomass samples, as previously published [5]. Extracted DNA was sequenced for analysis of bacterial diversity after amplifying the bacteria-specific V4-V5 region of the 16S rRNA gene using primers 518F (CCAGCAGCYGCGGTAAN) and 926R (CCGTCAATTCNTTTRAGT, CCGTCAATTTCTTTGAGT, CCGTCTATTCCTTTGANT) (https://vamps.mbl.edu/resources/primers.php). Sequencing was carried out as part of the Census of Deep Life initiative within the Deep Carbon Observatory at the Marine Biological Laboratory sequencing facility (https://www.mbl.edu/) on an Illumina MiSeq platform for amplicons.

## Bioinformatic and statistical analysis

Paired-end reads were imported and analyzed in RStudio [30] version 3.6 using the DADA2 package [31]. Quality check and trimming of the reads were performed (see deposited code for details on the parameter used). To optimize the merging of reads from the two separate libraries (the 2017 and 2018 datasets were sequenced in two different sequencing runs), we followed the DADA2 workflow for Big Data with little adaptation. Amplicon sequence variant (ASV) inference was performed on the dereplicated sequences after pooling samples and merging paired-end reads. Chimeric sequences were removed and prokaryotic taxonomy was assigned using a native implementation of the naiveBayesian classifier method against the silva database (v132; https://www.arb-silva.de/documentation/release-132/). ASVs were defined as clusters sharing 100% sequence identity [32]. All subsequent statistical analyses, data processing and plotting were carried out in the R statistical software version 3.6, using the phyloseq, vegan, ggtern and ggplot2 packages [33–36]. We obtained high-quality bacterial 16S rRNA gene libraries from 39 sediments and 35 fluid samples across the 48 hot spring sites. The count table, taxonomy assignment and phylogenetic tree were combined together with the environmental variables into a phyloseq object. Low prevalence ASVs (less than 5 reads), mitochondria, chloroplast-related sequences, common contaminants [37] and human pathogens were

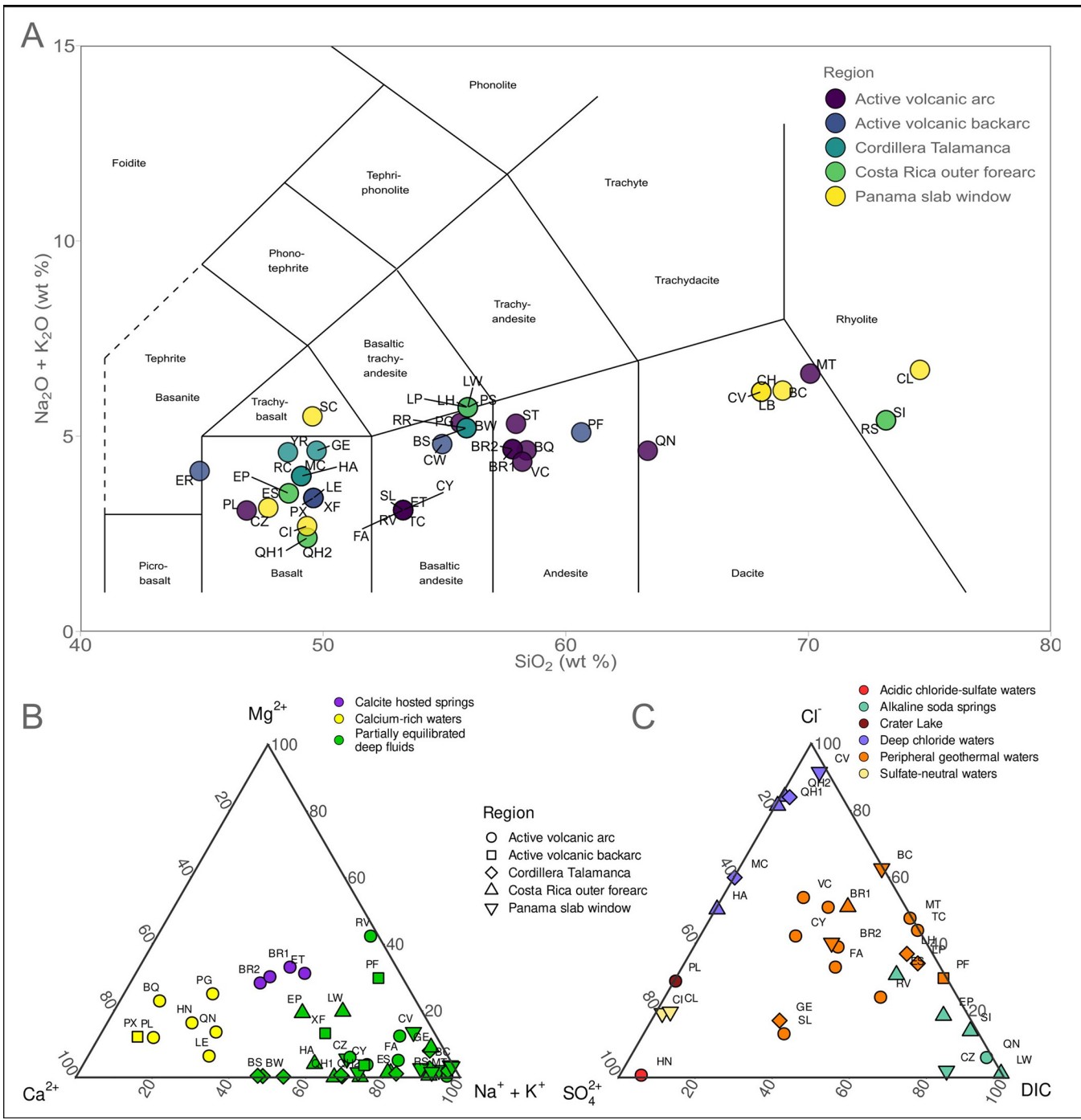

**Fig 2. Host rock classification and fluid geochemistry. A** Total alkali-silica (TAS) values showing a volcanic rock classification diagram [$(Na_2O+K_2O)$ vs. $SiO_2$ wt%] colored by the geographical area (classification according to Le Maitre et al. 1989 [28]). **B** and **C** Ternary diagrams showing the clustering of the fluid samples based on the major cations (B) and major anions (C).

removed from the dataset as described previously [38,39], removing 17.14% of the total reads. The ASV table was normalized to a common scale by transforming it to relative abundance within a sample and then multiplying this proportion by the median library size across all samples [40]. Statistical testing among variations in bacterial community composition was carried

out using the permanova analysis for the centroid of the groups (ADONIS) based on a distance matrix calculated using the Jensen-Shannon Divergence distance. Differences between groups were evaluated by One-way Analysis of Variance (One-way ANOVA) followed by Tukey's post hoc test. Non-metric MultiDimensional Scaling (nMDS) ordinations were used to identify geographical clustering and environmental explanatory variables using linear correlations of environmental vectors with the envfit function in vegan. The roles of different sampling factors in influencing the observed community patterns were tested using a permutation distance-based approach using the ANOSIM function of the vegan package. All p values were adjusted using the Holm correction for multiple hypothesis testing which reduces the probability of false positives. In order to test for the contribution of dispersal, we performed a correlation analysis using the Mantel test (Vegan package) between the beta-diversity matrix of the microbial composition and the pairwise distance (in km) between the sites. The test was repeated for the fluids and sediment samples separated and together setting the same site distance to 0 km.

## Results

### Geophysical parameters changing across the sampled section of the CAVA

Principal component analysis (PCA) of the geophysical parameters of 48 springs clusters them into several groups (Fig 1B). First, the eight southern Panama sites clearly separate from the rest of the dataset along the first axis of PCA (44.8% of the variation in the total dataset; Fig 1B, yellow markers). The Panama sites are distinguished from the rest of the sites by the presence of a significant slab window resulting in heavy input of volatiles from the mantle without volcanoes [7] (S1 Fig in S1 File). The rest of the sites span different parts of the subduction province (outer forearc, arc and backarc), but their geophysical attributes are also distinguished along-arc.

We grouped the sites into five subgroups based on subduction province, geophysical characteristics (PCA, Fig 1B), depth to slab (Fig 1C), and south/north regional classification: Costa Rica outer forearc (9 sites), Active volcanic arc (17 sites), Active volcanic backarc (5 sites), Cordillera Talamanca (9 sites), and Panama slab window (8 sites). These delineate both across-arc progression from outer forearc, to arc, to backarc, as well as along-arc progression from these subduction zone provinces to Cordillera Talamanca and Panama. Samples from the active volcanic arc areas exhibit higher temperatures (56.7 ± 17.5°C, with maximum of 88.9°C) than average (43.8 ± 15.9°C for all sites of the subduction zone), with lower values in the backarc (28.0°C), outer forearc (33.3±1.07°C), and Panama (36.0 ± 1.0°C). Conversely, pH values are the lowest in the active volcanic arc area (5.0 ± 1.9), with maximum values reported in the outer forearc of Costa Rica (9.0 ± 1.0) and in Panama (7.9 ± 1.2) (S2 Fig in S1 File).

### Geochemical characteristics

The diversity of host igneous rocks was obtained by analyzing elemental concentrations using the total alkali-silica (TAS) graph (Fig 2A). The sampled sites encompass a wide spectrum of volcanic host rock types, ranging from basalt to rhyolite. The variability in $SiO_2$ content among the sites classifies the majority of host rocks as basalts and andesites, with a smaller subset exhibiting a more dacitic and rhyolitic composition. The primary rock compositions do not exhibit a significant correlation with the subgroups identified above (Fig 2A).

Fluids were classified by their major anions [41] into: (i) acidic (pH 0 to 3) chloride-sulfate waters with direct input of magmatic gases (Fig 2C, red and burgundy dots), (ii) sulfate-poor, peripheral geothermal waters (orange dots), intermediate in composition between deeply-derived chloride-rich waters and soda springs (purple dots) characteristic of volcanic flanks

and forearcs (Fig 2C), and (iii) alkaline (pH 7 to 10) outer forearc sites poor in both sulfate and chloride (Fig 2C, blue dots), with the exception of sites CI and CL (Fig 2C, yellow dots) which contain high sulfate possibly from seawater influence. The majority of sites contain peripheral geothermal waters or alkaline springs, and only a few of them are deep chloride or acidic waters. Cation concentrations ($Mg^{2+}$, $Ca^{2+}$, and $Na^+$ plus $K^+$; organized according to Giggenbach [26]) differentiate between acidic arc volcanic sites (yellow dots) relatively enriched in $Ca^{2+}$, calcite hosted springs, and peripheral partially equilibrated waters. Flank geothermal sites and forearc springs (purple dots) define a trend away from the $Ca^{2+}$ apex, likely due to fluid neutralization accompanied by precipitation of calcite. Indeed, large travertine mounds were observed at many of these sites. The outer forearc as well as some forearc sites fall near the $Na^++K^+$ apex, suggesting that these were mature deep fluids partially equilibrated with alkali-feldspars and clays (S3 Fig in S1 File). All the other sites are mature deep sites (green dots). Many of the geological and geochemical variables show a strong degree of collinearity (S4 Fig in S1 File).

## Prokaryotic diversity and community composition

A total of 7,339,931 bacterial 16S rRNA gene amplicon reads comprising 36,534 amplicon sequence variants (ASVs) were obtained. Across all bacterial communities, the most abundant phylum is Proteobacteria (38.3 ± 22.6%), mainly including Gamma- (26.7 ± 22.1%) and Alpha-proteobacteria (6.0 ± 7.9%) (Fig 3). Other highly represented phyla include Bacteroidetes (11.8 ± 10.0%) and Chloroflexi (7.2 ± 6.8%). Aquificae has a high relative abundance only in a few sites from the active volcanic arc, with abundances up to 95% in PL and TC sediment samples and with lower abundances (7 to 48%) in 5 fluid samples of the volcanic arc with high temperatures. The Shannon richness index is significantly higher (ANOVA, $p < 0.001$) in sediments (4.6 ± 1.2) than in fluids (3.4 ± 1.2), and only 15.7% of the ASVs are shared between fluid and sediment samples (with 26.7% exclusive to fluids and 57.6% to sediments). Alpha diversity is not significantly different between regional groups for either sediments or fluids (S5 Fig in S1 File, ANOVA, p = ns) or between different hosting rock types as identified by TAS (ANOVA, p = ns). Bacterial community composition of sediment and fluid samples cluster into several groups across regions and sample types (Fig 3). A well-defined cluster of fluid samples from human-made wells spanning multiple geographical regions: Panama (i.e., CI, CZ, and CV), Cordillera Talamanca (i.e. BS, LP and LH) and backarc (i.e. CW, PX, and XF) were dominated by Firmicutes (with relative abundances ranging from 35% to 93%). In the non-well natural spring fluids, communities belonging to both fluid and sediment cluster into geographical groups with similar environmental (i.e., temperature and pH) and $^3He/^4He$ (reported as Rc/Ra values) (Fig 3 and S2 Fig in S1 File). Most of these sites from Panama (yellow sites) cluster together and have abundant Bacteroidetes, Deltaproteobacteria, and Alpha-proteobacteria. The active volcanic arc cluster (purple) contains a higher percentage of Aquificae, unidentified Proteobacteria, Chloroflexi, and unclassified phyla. The majority of the Cordillera Talamanca sites cluster together, with a similar phyla distribution to the others, but with a greater contribution from Proteobacteria. Costa Rica outer forearc sites cluster together, containing more Deltaproteobacteria, Nitrospirae, Chloroflexi, and Actinobacteria compared to other sites. The bottom set of clusters include the non-well sites from the backarc, but also sites from other regions. These clusters do not have dominating phyla driving their clustering, except for the absence of uncharacterized phyla. These clusters contain sites from similar geological regimes (Fig 3, colored dots on the left), but not similar temperatures, pH values, or $^3He/^4He$ ratios (Fig 3, heatmap values on the right).

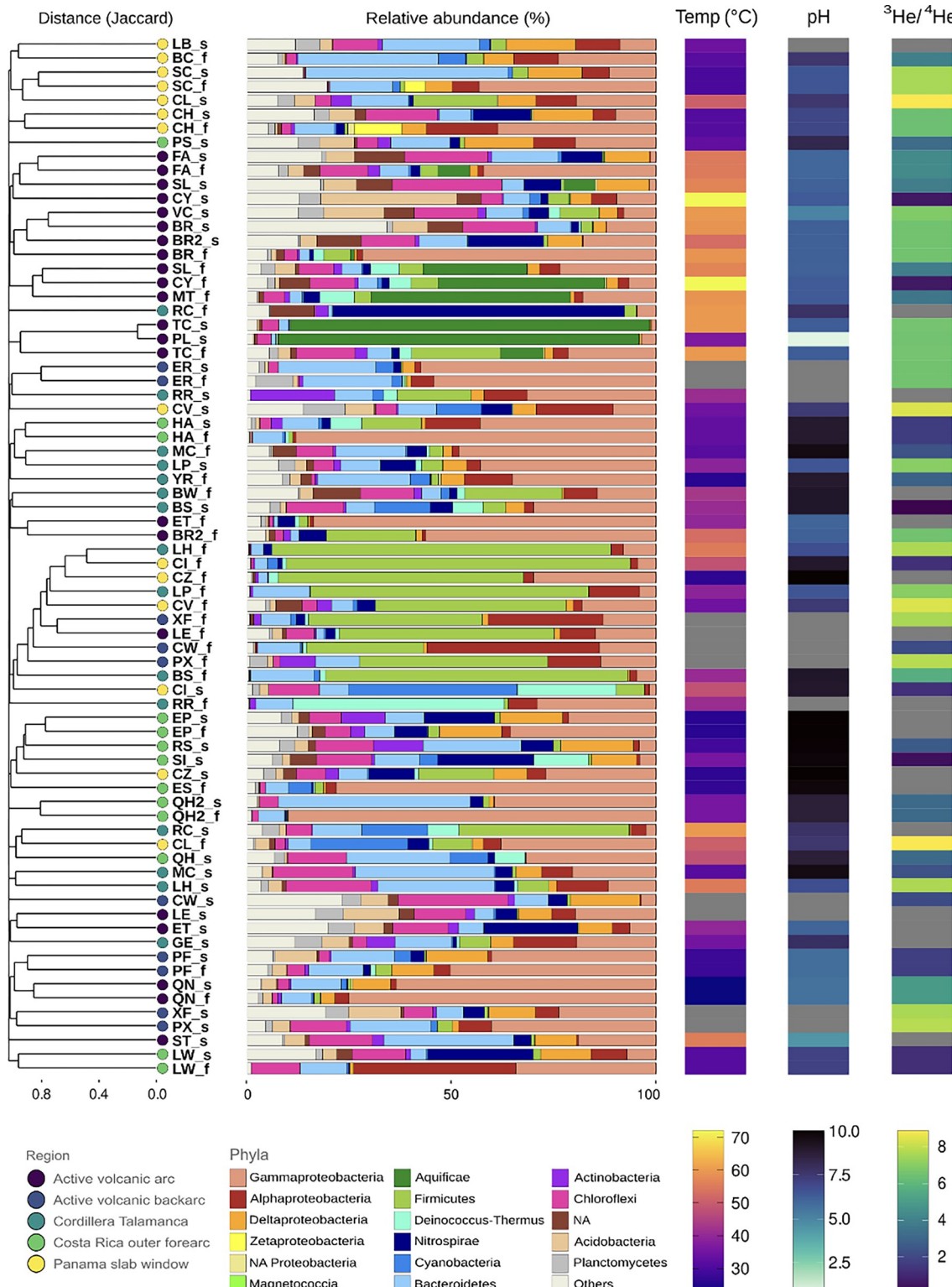

**Fig 3. Bacterial community composition, as shown by hierarchical clustering of Spearman Rank Coefficient based on 16S rRNA gene amplicon abundance.** On the left, cluster analysis based on Jaccard similarities (method complete), with site codes colored by geographical area; on the center, a bar plot showing prokaryotic community composition at the Phylum level and Class level for Proteobacteria only. "Others" includes taxa aggregated with an average relative abundance <1% across all samples, NA include the unidentified ASVs; on the right are environmental variables and geological settings: Temperature, pH, $^3He/^4He$; gray means no data.

A non-metric MultiDimensional Scaling (nMDS) ordination based on Jensen-Shannon Divergence (k = 3, stress = 0.176, best solution repeated 5 times) shows a significant separation between communities belonging to different along-arc geographical areas (Fig 4, S6-S8 Figs in S1 File). Here, we observe the same clusters that were observed in the hierarchical clustering from Panama, the volcanic arc and outer forearc regions (Fig 3), with a clear separation between outer forearc and volcanic arc sites compared to the rest of the samples. Sites belonging to the backarc and Cordillera Talamanca are not as well-separated from the other regions, in contrast to the result from the hierarchical clustering approach. Overall, these statistical analyses show shifts in the bacterial community structure along the CAVA convergent margin, with individual communities in each region being more similar to each other than to sites of other regions (Fig 4). The spatial autocorrelation analysis was not significant despite having a low p-value due to the large number of datapoints (S9 Fig in S1 File, Mantel test, r = 0.14, p < 0.01, n = 2071), suggesting that dispersal and site proximity are not driving the regional similarity observed. Microbial diversity is explained instead by region (ANOSIM, r = 0.38, p < 0.001) (Fig 4A–4C) and fluid geochemistry (ANOSIM, r = 0.38, p < 0.001) (Fig 4D and 4E).

Linear vector fitting against the nMDS ordination (on all 3 axis) reveals that temperature and pH do not explain bacterial diversity changes in our dataset when analyzing fluids and sediments together (envfit, p = ns, S2 Table in S1 File and S11 Fig in S1 File) or analyzing them separately. Despite this, temperature significantly correlates with nMDS axis 1 (N = 63, r = 0.77, p < 0.001) and pH correlates with nMDS axis 2 (N = 60, r = -0.51, p < 0.001). Microbial diversity changes do not correlate with mean annual precipitation in the area (envfit, p = ns, S2 Table in S1 File). This supports the observation that samples are deeply-sourced fluids, rather than recently deposited groundwaters, as $^{4}He$ to $^{20}Ne$ ratios show no air equilibration, and $^{3}He$ to $^{4}He$ ratios show significant input of mantle gasses (S10 Fig in S1 File) clearly separating our springs from shallow groundwaters. Linear vector fitting identifies $^{3}He/^{4}He$ as one of the key variables correlating with nMDS axis 3 (envfit, p < 0.01, S2 Table in S1 File), together with a number of covariates including several geophysical (envfit, p < 0.019, S2 Table in S1 File) and rock trace elements (Ni, Dy and Y, p < 0.01, S2 Table in S1 File). Bacterial community structure variations along nMDS axis 3 significantly correlate with $^{3}He/^{4}He$ (N = 56, r = -0.31, p < 0.01, Fig 4C).

## Discussion

Our sites span diverse geophysical, geological and geochemical regimes that cohesively vary along the CAVA: i) dip slab angle and subduction geometry, with steeper subduction in the north of Costa Rica; ii) changes in host rock geochemical composition and origin of volatiles, and iii) changes in geochemical and physicochemical parameters (like temperature, pH and water type) that are controlled by deep fluids and proximity to volcanic complexes. Collectively, these parameters support clear along-arc regional geological trends (Fig 1).

Our 16S rRNA gene amplicon-based bacterial community analyses from fluids and sediments of 48 hot springs over ~700 km of the CAVA show regional differentiation of bacterial diversity based on the geological features both across and along the arc (Fig 4). Geochemical parameters such as pH and temperature exert a primary control on community composition (Fig 4), in agreement with previous studies [12,21,42–44]. This reflects their direct effects on bacterial physiology, but also their indirect effects by determining rock weathering, element partitioning during water:rock interactions, redox state, and chemical speciation. However, temperature and pH alone do not describe the full variation we observe in the bacterial community composition of subsurface-derived fluids and the sediments they wash over. Most of

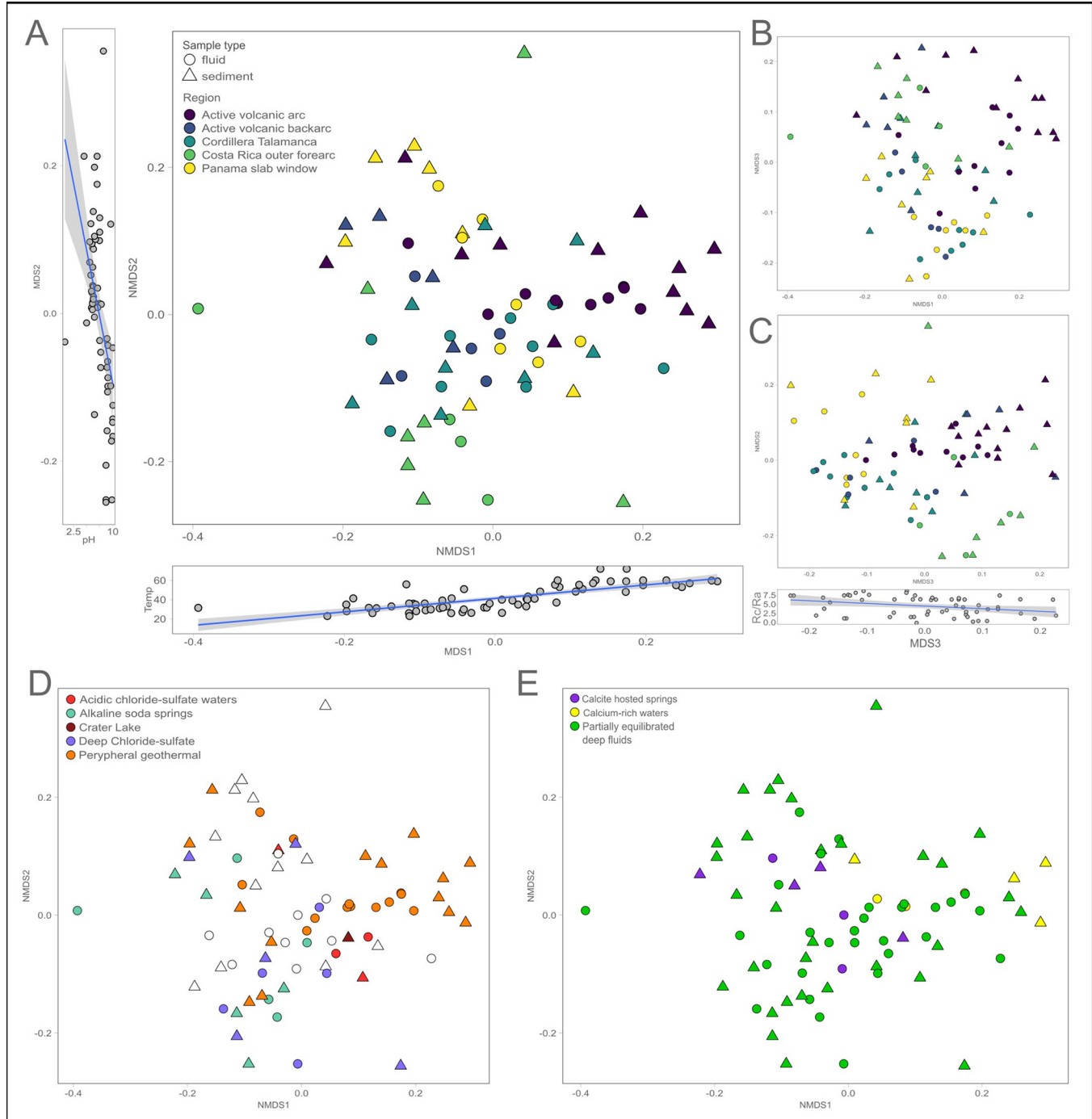

**Fig 4. Non-metric multidimensional scaling (nMDS) clustering of the sites based on bacterial community diversity and geochemical parameters. A** nMDS plot of the 16S rRNA gene amplicon bacterial diversity based on Jensen-Shannon Divergence measure in the fluids (circles) and sediments (triangles), coloured by geographical area. Flanking plots show the relationship between nMDS1 and temperature ($r^2$ = 0.77, p < 0.0001, n = 63) and nMDS2 and pH ($r^2$ = -0.50, p < 0.001, n = 60). **B** and **C**, the same nMDS plot showing nMDS1 and nMDS3 (B) and nMDS2 and nMDS3 (C) with the relationship between nMD3 and $^3He/^4He$ ($r^2$ = -0.31, p < 0.01, n = 56). **D** and **E**, the same nMDS plot in A, but colored according to the geochemistry-based grouping from anions (D) and cations (E). Empty symbols in D and E represent samples for which one or more ions were missing, and therefore have no placement in the ternary plot.

the microbial groups do not vary systematically with temperature and pH (Fig 3), suggesting that their distribution across the convergent margin is not a simple function of these two parameters [5,15]. Bacterial community composition also shifts across-arc with distance from the trench, progressing from the outer forearc, to the forearc, to the arc, following increasing distance from the subducting trench (DSub, Fig 3, S2 Table in S1 File). This agrees with previous findings from a subset of 18 of these sites (S1 Table in S1 File) that shifts in deep volatile delivery support changes in chemolithoautotrophic pathways and the redox reactions that power subsurface respiratory pathways across-arc [5,6,45].

In addition to this across-arc variation, the microbial diversity also shows clear shifts along the major geological provinces along the convergent margin. Microbial community structure from CAVA natural springs clearly differentiate by region (Fig 3). This regional pattern is upheld when populations in fluids or sediments are considered separately or together, suggesting that the freshly expressed fluids influence the microbial communities that collect at the seep orifice, where sediments are being continuously washed by the seeping fluids [23,24]. Separation is visible between microbial communities of the Costa Rica outer forearc, active volcanic arc, and Panama, whereas the Cordillera Talamanca and backarc sites show less strong regional clustering (Fig 4A, S6-S8 Figs in S1 File). Alpha diversity is not significantly different between these regions (S5 Fig in S1 File), suggesting that any regional differentiation between geological regimes is driven by the types of microbes present, rather than the total number of different microbes.

This regional differentiation of the microbial community is explained in our dataset using a combination of geochemical and geophysical information. The $^3$He/$^4$He ratio can be used as a proxy to disentangle the origin of the volatiles since the two isotopes have distinct origins with $^3$He being cosmogenic while $^4$He is produced radiogenically in Earth's crust [25,46]. Along the CAVA, the $^3$He/$^4$He ratios show varying degrees of mantle influence on the volatile origin (S1 Fig in S1 File). Microbial diversity significantly correlates with $^3$He/$^4$He values in our dataset (Fig 4C and S2 Table in S1 File), suggesting that the geological origin of the volatiles exerts a major control on community structure. This statement is further supported by the correlation of beta diversity with geophysical variables known to covary along the CAVA region (S2 Table in S1 File). Accordingly, the fluid geochemical composition reflects a strong influence of deep volatiles and interactions with the overriding crust, rather than in-mixing of surface-derived fluids (Fig 2). Together, the degree of hydrothermal maturation and crustal influence drive the differentiation of subsurface fluids into different geochemical compositions which influence the microbial communities at each site. The fluids cover the full range of variation in these properties, spanning calcite-rich waters, soda springs, acidic, deep chloride, and peripheral geothermal waters (Fig 2).

The fluids and sediments of the volcanic arc region have a higher relative abundance of Aquificae-related ASVs. Cultured representatives of this group are hydrogenotrophic chemolithoautotrophs with optimum growth temperatures between 65°C and 95°C [47,48], consistent with the highest temperatures for the area (Fig 3). The availability of hydrogen, together with the high temperatures, might favor the presence of these thermophilic hydrogen-oxidizing microorganisms.

The similarity between the Panama sites and the Active volcanic backarc sites cannot be explained by temperature and pH alone, but is instead best explained by similarities in aqueous geochemistry (Figs 3 and 4), which is ultimately controlled by the composition of the underlying bedrock, and the origin of volatiles. A recent study showed that the rock composition in Panama and in the backarc of Costa Rica is influenced by the Galapagos mantle plume [7], suggesting a common source for volatiles and trace elements used by microorganisms as important cofactors in metabolism [49]. An exception to these regional trends is a cluster of

mostly fluid samples from wells, which have a high abundance of Firmicutes (in particular Clostridiales). Members of the Firmicutes have previously been found in some of the deepest and oldest aquifers on Earth and other subsurface environments [50,51].

Besides the association of Aquificae with arc sites and Firmicutes with well fluids, no other single bacterial phylum shows clear regional trends. The observed trends are driven by changes at lower taxonomic levels than phyla. These changes are ultimately controlled by geological and geophysical parameters, significantly differentiating the microbial diversity in these contrasting regions. Given the collinearity of several of these variables with variations in latitude along CAVA (S4 Fig in S1 File), we considered whether the observed regional similarities are due to biogeographic patterns controlled by dispersal rather than the underlying geological features. Although dispersal can be an important driver of microbial community composition in hot springs [12] our results suggest that dispersal did not contribute significantly to changes in the bacterial composition at the large spatial scales of our study (S9 Fig in S1 File, Mantel test, r = 0.14, p < 0.01, N = 2071). The correlation between the similarity matrix and the pairwise geographic distance between the sites is weak in spite of the low p-value which is affected by the large number of observations [52], and driven by a low number of sites located in close proximity from each other (S9 Fig in S1 File).

Changes in the subsurface microbial diversity sampled through deeply-sourced seeps [23] have previously been related to changes in underlying geological and geophysical characteristics over smaller areas, such as northern Costa Rica [5,6], Peru [15] and Brothers volcano on the Kermadec arc [9], and similar large-scale differences in microbial diversity have been previously observed within single geological regimes in karst aquifers in Slovenia [53] and in hot springs in New Zealand [12], Yellowstone National Park [54,55] and South China [56]. Our study shows that subsurface microbial populations shift along large spatial scales (~700 Km) that encompass major shifts in the geology of the convergent margin. Even when sites from different geological regimes along the axis of the CAVA are physically close to each other, they have microbial communities clustering according to their geological characteristics, rather than proximity.

## Conclusion

Our study shows significant changes in the subsurface microbial community composition along the CAVA. The observed regional differentiation is not shaped by any single environmental variable or phylum. Instead, it appears to be driven by the complex interplay of conditions and microbial communities present in each geographical region. Even though each region has similar ranges of pH and temperature, we suggest that regionality in microbial community composition is affected by: (i) direct magmatic degassing and mature hydrothermal activity in the Costa Rican active volcanic zone, as shown by helium and carbon isotopes [11], (ii) volatiles derived from the slab during subduction in the outer forearc, notably oxidized aqueous sulfur [6], (iii) volatiles from flat slab subduction in the Cordillera Talamanca, where the slab remains shallow well into its interaction with the mantle [57], and (iv) the absence of a slab in Panama, where the only volatiles are directly derived from the mantle and crust [7]. These factors ultimately influence the ecological niches available in the subsurface, controlling the community structure, composition and functions of the subsurface biosphere. Our work demonstrates that coupling between deep Earth processes and surface manifestation of subsurface activities [58] helps shape the microbial communities of subsurface ecosystems. Since previous results suggest that these microbial communities play a significant role in mediating the volatile cycling between the Earth interior and its surface [5,6,8,15,59], we conclude that these

biology-geology feedbacks extend to the large along-arc spatial scale spanning major changes in the geological setting.

## Supporting information

**S1 File. All supporting information, figures (S1-S11 Figs), and tables (S1 and S2 Tables) are included together as a single separate file.**
(DOCX)

## Author Contributions

**Conceptualization:** J. Maarten de Moor, Peter H. Barry, Karen G. Lloyd, Donato Giovannelli.

**Data curation:** Marco Basili, Timothy J. Rogers, Mustafa Yücel, J. Maarten de Moor, Peter H. Barry, Matthew O. Schrenk, Carlos J. Ramirez, Angelina Cordone, Karen G. Lloyd, Donato Giovannelli.

**Formal analysis:** Marco Basili, Timothy J. Rogers, Mustafa Yücel, J. Maarten de Moor, Peter H. Barry, Gerdhard L. Jessen, Ricardo Sánchez-Murillo, Sabin Zahirovic, Deborah Bastoni, Angelina Cordone, Karen G. Lloyd, Donato Giovannelli.

**Funding acquisition:** J. Maarten de Moor, Peter H. Barry, Gerdhard L. Jessen, Karen G. Lloyd, Donato Giovannelli.

**Investigation:** Marco Basili, Timothy J. Rogers, Mayuko Nakagawa, J. Maarten de Moor, Peter H. Barry, Matthew O. Schrenk, Gerdhard L. Jessen, Ricardo Sánchez-Murillo, Sabin Zahirovic, David V. Bekaert, Carlos J. Ramirez, Karen G. Lloyd, Donato Giovannelli.

**Methodology:** Mayuko Nakagawa, Mustafa Yücel, J. Maarten de Moor, Peter H. Barry, Gerdhard L. Jessen, Sabin Zahirovic, Carlos J. Ramirez, Karen G. Lloyd, Donato Giovannelli.

**Project administration:** J. Maarten de Moor, Peter H. Barry, Gerdhard L. Jessen, Karen G. Lloyd, Donato Giovannelli.

**Resources:** J. Maarten de Moor, Peter H. Barry, Karen G. Lloyd, Donato Giovannelli.

**Software:** Sabin Zahirovic, Donato Giovannelli.

**Supervision:** J. Maarten de Moor, Peter H. Barry, Karen G. Lloyd, Donato Giovannelli.

**Validation:** J. Maarten de Moor, Peter H. Barry, Karen G. Lloyd, Donato Giovannelli.

**Visualization:** Marco Basili, Deborah Bastoni.

**Writing – original draft:** Marco Basili, J. Maarten de Moor, Peter H. Barry, Karen G. Lloyd.

**Writing – review & editing:** Marco Basili, Timothy J. Rogers, Mayuko Nakagawa, Mustafa Yücel, J. Maarten de Moor, Peter H. Barry, Matthew O. Schrenk, Gerdhard L. Jessen, Ricardo Sánchez-Murillo, Sabin Zahirovic, David V. Bekaert, Carlos J. Ramirez, Deborah Bastoni, Angelina Cordone, Karen G. Lloyd.

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
