## [Editor Report · Decision Letter 0]

20 Feb 2024

PONE-D-24-05844Subsurface microbial community structure shifts along the geological features of the Central American Volcanic ArcPLOS ONE

Dear Dr. Lloyd,

Thank you for submitting your manuscript to PLOS ONE. I have carefully reviewed your work and believe it has the potential to contribute significantly to our journal. However, before we can proceed with the peer review process, I would like to request some modifications to enhance the clarity and structure of your manuscript.

1. Please ensure that the title of your article matches identically in both the PLOS ONE login system and the original manuscript. Consistency in the title is crucial for proper indexing and accessibility.

2. Kindly add continuous line numbers to your manuscript. This will assist both the reviewers and editors in providing specific feedback and navigating through the document more efficiently.

3. Rearrange the manuscript so that the Material and Methodology section precedes the Results section. This adjustment will align your manuscript with the standard structure expected in academic publications.

4. Include a conclusion section after the discussion. A well-defined conclusion is essential to summarize key findings and provide a clear resolution to the research questions or objectives.

Please address these modifications in your revised manuscript and resubmit it. I believe that these changes will strengthen the overall quality of your submission.

We look forward to receiving your revised manuscript.

Kind regards,

Rizwan Sarwar Awan

Academic Editor

PLOS ONE

Journal Requirements:

2. In your Methods section, please provide additional information regarding the permits you obtained for the work. Please ensure you have included the full name of the authority that approved the field site access and, if no permits were required, a brief statement explaining why

"The authors acknowledge the Biology Meets Subduction Project, funded by the Alfred P. Sloan Foundation and the Deep Carbon Observatory (G-2016-7206) to P.H.B,J.M.d.M, D.G., and K.G.L., with DNA sequencing from the Census of Deep Life"

"The authors acknowledge the Biology Meets Subduction Project, funded by the Alfred P. Sloan Foundation and the Deep Carbon Observatory (G-2016-7206) to P.H.B, J.M.d.M, D.G., and K.G.L., with DNA sequencing from the Census of Deep Life. Additional support came from NSF FRES (Award# 21211637) to P.H.B., J.M.d.M and K.G.L. U. S. Department of Energy, Office of Science, Office of Biological and Environmental Research, Genomic Science Program (DE-SC0020369 to K.G.L). FONDECYT Grant 11191138 and COPAS COASTAL ANID FB210021 (ANID Chile) to G.L.J. D.G. was partially supported by funding from the European Research Council (ERC) under the European Union’s Horizon 2020 research and innovation program Grant Agreement No. 948972—COEVOLVE—ERC-2020-STG. M.B. was funded by the EU CampusWorld scholarship from UNIVPM to visit the laboratory of K.G.L. in the framework of a research collaboration between K.G.L. and D.G. S.Z. was supported by Australian Research Council grant DE210100084, and Alfred P Sloan grants G-2017-9997 and G-2018-11296. GPlates development is funded by the AuScope National Collaborative Research Infrastructure System (NCRIS) program."

---

## [Author Response · Author response to Decision Letter 0]

5 Apr 2024

Dear Rizwan Sarwar Awan, 

Thank you for your editorial and journal revision suggestions. We have uploaded our revised documents and provide point by point responses below, in italics.

PONE-D-24-05844

Subsurface microbial community structure shifts along the geological features of the Central American Volcanic Arc

PLOS ONE

Dear Dr. Lloyd,

Thank you for submitting your manuscript to PLOS ONE. I have carefully reviewed your work and believe it has the potential to contribute significantly to our journal. However, before we can proceed with the peer review process, I would like to request some modifications to enhance the clarity and structure of your manuscript.

1. Please ensure that the title of your article matches identically in both the PLOS ONE login system and the original manuscript. Consistency in the title is crucial for proper indexing and accessibility.

Done.

2. Kindly add continuous line numbers to your manuscript. This will assist both the reviewers and editors in providing specific feedback and navigating through the document more efficiently.

Done.

3. Rearrange the manuscript so that the Material and Methodology section precedes the Results section. This adjustment will align your manuscript with the standard structure expected in academic publications.

Done.

4. Include a conclusion section after the discussion. A well-defined conclusion is essential to summarize key findings and provide a clear resolution to the research questions or objectives.

Done.

Please address these modifications in your revised manuscript and resubmit it. I believe that these changes will strengthen the overall quality of your submission.

This is the current document.

This has been uploaded.

This has been uploaded.

We look forward to receiving your revised manuscript.

Kind regards,

Rizwan Sarwar Awan

Academic Editor

PLOS ONE

Journal Requirements:

2. In your Methods section, please provide additional information regarding the permits you obtained for the work. Please ensure you have included the full name of the authority that approved the field site access and, if no permits were required, a brief statement explaining why

This has been added.

"The authors acknowledge the Biology Meets Subduction Project, funded by the Alfred P. Sloan Foundation and the Deep Carbon Observatory (G-2016-7206) to P.H.B,J.M.d.M, D.G., and K.G.L., with DNA sequencing from the Census of Deep Life"

"The authors acknowledge the Biology Meets Subduction Project, funded by the Alfred P. Sloan Foundation and the Deep Carbon Observatory (G-2016-7206) to P.H.B, J.M.d.M, D.G., and K.G.L., with DNA sequencing from the Census of Deep Life. Additional support came from NSF FRES (Award# 21211637) to P.H.B., J.M.d.M and K.G.L. U. S. Department of Energy, Office of Science, Office of Biological and Environmental Research, Genomic Science Program (DE-SC0020369 to K.G.L). FONDECYT Grant 11191138 and COPAS COASTAL ANID FB210021 (ANID Chile) to G.L.J. D.G. was partially supported by funding from the European Research Council (ERC) under the European Union’s Horizon 2020 research and innovation program Grant Agreement No. 948972—COEVOLVE—ERC-2020-STG. M.B. was funded by the EU CampusWorld scholarship from UNIVPM to visit the laboratory of K.G.L. in the framework of a research collaboration between K.G.L. and D.G. S.Z. was supported by Australian Research Council grant DE210100084, and Alfred P Sloan grants G-2017-9997 and G-2018-11296. GPlates development is funded by the AuScope National Collaborative Research Infrastructure System (NCRIS) program."

This has been changed.

No IRB was required for this study as it did not include human subjects.

---

## [Decision Letter · Decision Letter 1]

13 May 2024

PONE-D-24-05844R1Subsurface microbial community structure shifts along the geological features of the Central American Volcanic ArcPLOS ONE

Dear Dr. Lloyd,

Thank you for submitting your manuscript to PLOS ONE. After careful consideration, we feel that it has merit but does not fully meet PLOS ONE’s publication criteria as it currently stands. Therefore, we invite you to submit a revised version of the manuscript that addresses the points raised during the review process.

**The review process has now been completed for your manuscript, the reviewers have raised critical comments that you need to pay attention and revise your work accordingly.**

We look forward to receiving your revised manuscript.

Kind regards,

Rizwan Sarwar Awan

Academic Editor

PLOS ONE

Journal Requirements:

Reviewers' comments:

Reviewer's Responses to Questions

**Comments to the Author**

1. If the authors have adequately addressed your comments raised in a previous round of review and you feel that this manuscript is now acceptable for publication, you may indicate that here to bypass the “Comments to the Author” section, enter your conflict of interest statement in the “Confidential to Editor” section, and submit your "Accept" recommendation.

Reviewer #1: (No Response)

Reviewer #2: All comments have been addressed

2. Is the manuscript technically sound, and do the data support the conclusions?

Reviewer #1: Yes

Reviewer #2: Yes

3. Has the statistical analysis been performed appropriately and rigorously? 

Reviewer #1: Yes

Reviewer #2: Yes

4. Have the authors made all data underlying the findings in their manuscript fully available?

Reviewer #1: Yes

Reviewer #2: Yes

5. Is the manuscript presented in an intelligible fashion and written in standard English?

Reviewer #1: No

Reviewer #2: Yes

6. Review Comments to the Author

**Reviewer #1**: Basili et al. built upon previous understandings of across-arc microbial community variations in the Central American Volcanic Arc and investigated the microbial community variations *along* the convergent margin. The study is a useful addition to our understanding of biosphere-geosphere interactions. The study appears to be technically sound and associated methods are described in sufficient details. However, the current presentation is somewhat difficult to follow especially when reading for the first time. I suggest the following improvements:

1. Please clarify the relationship between the microbial sample set in this manuscript and in other companion papers, e.g. Fullerton et al. 2021 Nat. Geosci.: Were any of the microbial samples used in this study published before? If so, please specify which one(s) were downloaded data and add citations; if not, please describe whether or how the sampling/sample processing/resulting data differ between samples that appear to have the same names across papers.

2. I suggest clarifying what direction is meant by "across-arc" and "along arc" in earlier sections of the manuscript. It might be helpful to add arrows to Fig.1 to help readers comprehend along which direction are you analyzing microbial variations. Currently, the first instance where this is clarified seems to be in the Results section, line 198-199.

3. Lines 196-197: Please make the naming of these subgroups of regions consistent with those in Figs 1A, S1, etc. Please also add the number of samples per region to the Figures where appropriate.

4. Lines 222-226 & Fig. 2B&C: Can you map the regions (arc, forearc, backarc, Panama slab window, etc.) onto Fig. 2B&C? e.g. by using different shapes? It is hard to follow the descriptions here for readers unfamiliar with which samples were from "outer forearc" or "forearc" or "flank geothermal sites". Same for Fig. S3.

5. Lines 277-279 & Fig. S11: Fig S11 is hard to interpret and doesn't seem to support what is said here based on its current presentation. Please map different regions and sample types onto Fig S11 by using different colors, shapes, etc.

6. Line 266-269 & Fig. 3: The interpretation of the hierarchical clustering is not quite straightforward. It looks like most sample pairs have Jaccard dist. close to 1, so it is hard to see many obvious "clusters". What is the rationale behind choosing Jaccard distance? Have you considered/tried other distance metrics?

Other minor issues:

- Line 103: the word "active" is duplicated.

- Line 252: by "dark blue" do you mean purple?

- Line 262: Based on Jensen-Shannon Divergence or Jaccard distance? The caption of Fig 4 says Jaccard distance.

- Line 302: Please cite the figure(s) that support what you said here.

- Line 346: Please cite the figure(s) that support what you said here.

- Line 346: by "Costa Rica backarc" do you mean Costa Rica outer forearc?

- Fig S4: Please explain what is meant by the dendrogram on the left side of the heatmap.

**Reviewer #2: **This manuscript is a very interesting paper, which presents microbial community changes across the Central American Volcanic Arc, and discusses underling mechanisms controlling the shifts in microbial composition in each sites along the arc. The paper is well written, and the results support their conclusion overall. As there are some many data and parameters considered, the authors need to revise their paper by presenting their discovery and point of views more clearly. My major concerns are:

(1) The authors suggest that bacterial community composition shifts across-arc with distance from the trench in addition to changes in pH and temperature, without giving further explanation as to what kind of change leads to such shifts in bacterial community (Line 299-314). They gave an interpretation that such changes in sampling location relative to the cross-arc may result in a shift in deep volatile delivery, without further evidence to support this notion. In their discussion part, the authors need to give evidence to support and explain this clearly.

(2)

Another main point of the manuscript is that microbial community composition shifts significantly according to geological changes along the CAVA, supporting the fundamental role that tectonic processes play in shaping subsurface microbial ecosystems. Again the authors need to explain more about this geological change. From their Line 315-354, I tried to guess that such geological change include deep volatile and bedrock composition, which I am not so sure. This is because the formal (volatile) has interacted with overriding crust, through which to change fluid geochemical composition. What is the relationship between overriding crust and bedrock? Are they the same stuff? The authors need to revise their discussion and present their notions more clearly. Also If geological change has something to do with deep volatile, then this has been mentioned in their Line 312-314. If the two paragraphs were discussing the same thing, the authors can try to revise the contents concisely.

Details of the sampled section need to be added.

Line 59-61: Rephrase.

Line 63: Be specific of what “geological setting” means?

Line 72-73: Previous sentence did not mention that scientist names. Thus, I cannot understand what these authors mean.

Line 78: along-arc differences of what?

Line 101-104: Rephrase.

Line 103: Delete one “active”.

Line 126: Detailed methods of what tests?

Line 309: Please be specific of what kind of change in bacterial community have changed across-arc.

Line 311-314: I did not get the logic here. How does the author come up with this conclusion based on previous discussion?

Line 316: What do you mean by geological change? Bedrock composition and volatile?

Line 246 Rephrase. The similarity in microbial community between these two sites?

Figure and figure captions

Fig. 4: font size within figure are too small to read.

7. PLOS authors have the option to publish the peer review history of their article (what does this mean?). If published, this will include your full peer review and any attached files.

Reviewer #1: No

Reviewer #2: No

---

## [Author Response · Author response to Decision Letter 1]

7 Jul 2024

We thank the reviewers for their helpful comments. Please find our point by point responses to all of them below in italics.

Reviewer #1: Basili et al. built upon previous understandings of across-arc microbial community variations in the Central American Volcanic Arc and investigated the microbial community variations *along* the convergent margin. The study is a useful addition to our understanding of biosphere-geosphere interactions. The study appears to be technically sound and associated methods are described in sufficient details. However, the current presentation is somewhat difficult to follow especially when reading for the first time. I suggest the following improvements:

1. Please clarify the relationship between the microbial sample set in this manuscript and in other companion papers, e.g. Fullerton et al. 2021 Nat. Geosci.: Were any of the microbial samples used in this study published before? If so, please specify which one(s) were downloaded data and add citations; if not, please describe whether or how the sampling/sample processing/resulting data differ between samples that appear to have the same names across papers.

Response: We thank the reviewer for catching this oversight. The full sample IDs for each site are now included in Table S1, with the following information in the title “Location and physico-chemical parameters of the sampled sites. Sample IDs that have numbers starting with 17 are sites where 16S rRNA gene amplicon data was published in Fullerton et al., 2021.” 

2. I suggest clarifying what direction is meant by "across-arc" and "along arc" in earlier sections of the manuscript. It might be helpful to add arrows to Fig.1 to help readers comprehend along which direction are you analyzing microbial variations. Currently, the first instance where this is clarified seems to be in the Results section, line 198-199.

Response: The following text has been added to the introduction Across-arc refers to moving from the trench, to the outer forearc, to the forearc, to the arc, while along-arc refers to moving into different sections of the convergent margin along its axis (Fig. 1 arrows). And clarifying arrows have been added to the figure, with an explanation in the figure caption. 

3. Lines 196-197: Please make the naming of these subgroups of regions consistent with those in Figs 1A, S1, etc. Please also add the number of samples per region to the Figures where appropriate.

Response: The text has been changed to match the region names in the figure. It now reads “Costa Rica outer forearc (9 sites), Active volcanic arc (17 sites), Active volcanic backarc (5 sites), Cordillera Talamanca (9 sites), and Panama slab window (8 sites).” Adding the number of samples per region to this already-busy figure might overwhelm it. One can get a visual idea of the number of sites in each region by looking at the figure and then read in the text to see the real numbers.

4. Lines 222-226 & Fig. 2B&C: Can you map the regions (arc, forearc, backarc, Panama slab window, etc.) onto Fig. 2B&C? e.g. by using different shapes? It is hard to follow the descriptions here for readers unfamiliar with which samples were from "outer forearc" or "forearc" or "flank geothermal sites". Same for Fig. S3.

Response: Regions have been mapped to the figure as requested.

5. Lines 277-279 & Fig. S11: Fig S11 is hard to interpret and doesn't seem to support what is said here based on its current presentation. Please map different regions and sample types onto Fig S11 by using different colors, shapes, etc.

Response: Fixed.

6. Line 266-269 & Fig. 3: The interpretation of the hierarchical clustering is not quite straightforward. It looks like most sample pairs have Jaccard dist. close to 1, so it is hard to see many obvious "clusters". What is the rationale behind choosing Jaccard distance? Have you considered/tried other distance metrics?

Response: We have tested several metrics (manhattan, bray-curtis, jaccard and hellinger) and, keeping in mind the specific strenght and weaknesses of every metrics decided to show the Jaccard based distance, since it is more robust for diversity data. While the jaccard distance is high, several well formed cluster are formed fro jaccard distances below 0.8. We discuss these clusters in the manuscript.

Other minor issues:

- Line 103: the word "active" is duplicated.

Response: fixed.

- Line 252: by "dark blue" do you mean purple?

Response: Yes, this is now fixed.

- Line 262: Based on Jensen-Shannon Divergence or Jaccard distance? The caption of Fig 4 says 

Response: Thank you for spotting the error. We corrected the caption reporting the JSD as the distance.

- Line 302: Please cite the figure(s) that support what you said here.

Response: A reference to Fig. 4 has now been added.

- Line 346: Please cite the figure(s) that support what you said here.

Response: References to Figs. 3 and 4 have now been added.

- Line 346: by "Costa Rica backarc" do you mean Costa Rica outer forearc?

Response: This should mean “Active volcanic backarc” and has now been changed.

- Fig S4: Please explain what is meant by the dendrogram on the left side of the heatmap.

Response; The following text has been added to the figure legend: “The dendrogram on the left side of the heatmap shows clustering based on similarity of distributions of that geochemical variable across sites.”

Reviewer #2: This manuscript is a very interesting paper, which presents microbial community changes across the Central American Volcanic Arc, and discusses underling mechanisms controlling the shifts in microbial composition in each sites along the arc. The paper is well written, and the results support their conclusion overall. As there are some many data and parameters considered, the authors need to revise their paper by presenting their discovery and point of views more clearly. My major concerns are:

(1) The authors suggest that bacterial community composition shifts across-arc with distance from the trench in addition to changes in pH and temperature, without giving further explanation as to what kind of change leads to such shifts in bacterial community (Line 299-314). They gave an interpretation that such changes in sampling location relative to the cross-arc may result in a shift in deep volatile delivery, without further evidence to support this notion. In their discussion part, the authors need to give evidence to support and explain this clearly.

Response: There is a large body of geochemical literature studying shifts in the volatiles emitted between these regions (CO2, S, and others – availability of sulfur controls iron availability). To reduce this to a few variables is not our purpose here. Instead, we are looking at how the microbial community changes along with the different geological regimes with different composite properties that would be inappropriate to pick apart, since the microbes experience all of them, rather than just one or two. To help clarify this, we have add some specific elements involved in the volatile cycling in the introduction here: 

Definition of the volatiles:

Subduction zones are the primary tectonic settings that transfer volatiles between Earth’s surface and subsurface1,2. Many of these volatiles, such as inorganic carbon and redox-active elements like carbon, nitrogen, sulfur, and iron compounds, are biologically reactive and can be used for biomass synthesis and energy production.

Description of how they change along this particular arc:

Costa Rica and Panama have complex geological settings due to the intersection of several tectonic plates (Fig. 1a; Caribbean, Cocos, Nazca, and the Panama microplate) and the Cocos Ridge15. This complex tectonic setting creates regional-scale along-arc differences in volatile fluxes. The most prominent tectonic features in the area are: a) the Middle American Trench; b) the Cocos Ridge, a submarine volcanic range formed by the Galapagos hot spot track that is subducted in southern Costa Rica; and c) the Panama Fracture Zone, a transform fault zone that defines the triple junction between the Cocos, Nazca, and Caribbean plates off the coast of southern Costa Rica and northern Panama. Costa Rica is traversed by four mountain ranges, with most volcanoes located in the Guanacaste and Central volcanic ranges16,17. Central-Northern Costa Rica is characterized by basaltic-andesitic volcanism with multiple volcanic features such as thermal waters, acidic rivers or streams rich in sulfur, iron, and silicates16. In contrast to Costa Rica, Panama is characterized by a major left-lateral transform fault, along which the Nazca Plate is moving eastwards and is subducting beneath Colombia. This results in a slab window, where a break in the subducting slab allows for a greater mantle influence in the area15. The occurrence of this slab window notably explains the absence of arc volcanism and the relatively low seismic activity in Western Panama15,18. 

(2)

Another main point of the manuscript is that microbial community composition shifts significantly according to geological changes along the CAVA, supporting the fundamental role that tectonic processes play in shaping subsurface microbial ecosystems. Again the authors need to explain more about this geological change. From their Line 315-354, I tried to guess that such geological change include deep volatile and bedrock composition, which I am not so sure. This is because the formal (volatile) has interacted with overriding crust, through which to change fluid geochemical composition. What is the relationship between overriding crust and bedrock? Are they the same stuff? The authors need to revise their discussion and present their notions more clearly. Also If geological change has something to do with deep volatile, then this has been mentioned in their Line 312-314. If the two paragraphs were discussing the same thing, the authors can try to revise the contents concisely.

Response: We have included a dedicated section in the introduction describing briefly how the different volatiles and geological parameters change along the CAVA, and included relevant references both in the introduction and the discussion. Such changes are well documented in the literature, and a thorough description is beyond the scope of this manuscript. 

Details of the sampled section need to be added.

Line 59-61: Rephrase.

Response: This has been changed to: “Previous across-arc work shows that subsurface chemosynthesis-based communities vary by fluid sources and upper plate processes (e.g., 6,7).”

Line 63: Be specific of what “geological setting” means?

Response: The term “geological setting” was used on purpose to refer to a combination of all the variables that can change in subsurface environments. The lack of specificity is intentional so as not to elevate any individual variable (e.g., rock type, porosity, mineralogy, rainfall, altitude, geological landform, etc.) but rather that the combination of all these things were found in previous work to affect the microbial community composition.

Line 72-73: Previous sentence did not mention that scientist names. Thus, I cannot understand what these authors mean.

Response: the reference has been changed to “this study”.

Line 78: along-arc differences of what?

Response: this has been changed to “This complex tectonic setting creates regional-scale along-arc differences in volatile fluxes.”

Line 101-104: Rephrase.

Response: This has been rewritten at “Furthermore, deep fluid communities, which have been shown to be from deep sources because they have relatively high 3He/4He suggesting significant contributions from mantle volatiles and other geochemical indicators of active hydrothermal activity (e.g., hydrothermally derived anions and cations)7,23–25, do not overlap with the microbial communities in the surrounding soils24.”

Line 103: Delete one “active”.

Response: Fixed.

Line 126: Detailed methods of what tests?

Response: Detailed methods for geochemical measurements and limits of detection were previously reported5.

Line 309: Please be specific of what kind of change in bacterial community have changed across-arc.

Response: We previously published two papers going into detail about what bacterial changes occur across-arc for one half of this dataset (Fullerton et al., 2021 and Rogers et al., 2022). In the current paper, we don’t want to recapitulate those previous findings beyond simply confirming the trend in total microbial community compositions is upheld when we include the expanded dataset.

Line 311-314: I did not get the logic here. How does the author come up with this conclusion based on previous discussion?

Response: This is related to the above response. Does this change in the text clarify it? “Bacterial community composition also shifts across-arc with distance from the trench, progressing from the outer forearc, to the forearc, to the arc, following increasing distance from the subducting trench (DSub, Fig 3, Supplementary Table 2). This agrees with previous findings from a subset of 18 of these sites (Table S1) that shifts in deep volatile delivery support changes in chemolithoautotrophic pathways and the redox reactions that power subsurface respiratory pathways across-arc5,6,31.”

Line 316: What do you mean by geological change? Bedrock composition and volatile?

Response: We mean the countless variables that change in different geological provinces. We have changed it to say: “In addition to this across-arc variation, the microbial diversity also shows clear shifts along the major geological provinces along the convergent margin.”

Line 246 Rephrase. The similarity in microbial community between these two sites?

Response: This has been changed to “A well-defined cluster of fluid samples from human-made wells spanning multiple geographical regions: Panama (i.e., CI, CZ, and CV), Cordillera Talamanca (i.e. BS, LP, and LH) and backarc (i.e. CW, PX, and XF) were dominated by Firmicutes (with relative abundances ranging from 35 % to 93 %)”

Figure and figure captions

Fig. 4: font size within figure are too small to read.

Response: Font size has been adjusted so that a full size figure when printed in the final pdf can be readable.

---

## [Decision Letter · Decision Letter 2]

30 Jul 2024

Subsurface microbial community structure shifts along the geological features of the Central American Volcanic Arc

PONE-D-24-05844R2

Dear Dr. Karen Lloyd,

We’re pleased to inform you that your manuscript has been judged scientifically suitable for publication and will be formally accepted for publication once it meets all outstanding technical requirements.

Kind regards,

Rizwan Sarwar Awan

Academic Editor

PLOS ONE

Additional Editor Comments (optional):

Reviewers' comments:

Reviewer's Responses to Questions

**Comments to the Author**

1. If the authors have adequately addressed your comments raised in a previous round of review and you feel that this manuscript is now acceptable for publication, you may indicate that here to bypass the “Comments to the Author” section, enter your conflict of interest statement in the “Confidential to Editor” section, and submit your "Accept" recommendation.

Reviewer #1: All comments have been addressed

Reviewer #2: All comments have been addressed

2. Is the manuscript technically sound, and do the data support the conclusions?

Reviewer #1: (No Response)

Reviewer #2: Yes

3. Has the statistical analysis been performed appropriately and rigorously? 

Reviewer #1: (No Response)

Reviewer #2: Yes

4. Have the authors made all data underlying the findings in their manuscript fully available?

Reviewer #1: (No Response)

Reviewer #2: Yes

5. Is the manuscript presented in an intelligible fashion and written in standard English?

Reviewer #1: (No Response)

Reviewer #2: Yes

6. Review Comments to the Author

Reviewer #1: (No Response)

Reviewer #2: (No Response)

7. PLOS authors have the option to publish the peer review history of their article (what does this mean?). If published, this will include your full peer review and any attached files.

Reviewer #1: No

Reviewer #2: No

---

## [Editor Report · Acceptance letter]

12 Sep 2024

PONE-D-24-05844R2 

PLOS ONE

Dear Dr. Lloyd, 

I'm pleased to inform you that your manuscript has been deemed suitable for publication in PLOS ONE. Congratulations! Your manuscript is now being handed over to our production team.

Kind regards, 

on behalf of

Dr. Rizwan Sarwar Awan 

Academic Editor

PLOS ONE